# Multi-Tier Heterogeneous Beam Management for Future Indoor FSO Networks

**Michael B. Rahaim** [1,*], **Thomas D. C. Little** [2] **and Mona Hella** [3]

1   Engineering Department, University of Massachusetts, Boston, MA 02125, USA
2   ECE Department, Boston University, Boston, MA 02215, USA; tdcl@bu.edu
3   ECE Department, Rensselaer Polytechnic Institute, Troy, NY 12180, USA; hellam@rpi.edu
*   Correspondence: Michael.Rahaim@umb.edu

**Featured Application: Wireless Communications and Networking.**

**Abstract:** To meet the growing demand for wireless capacity, communications in the Terahertz (THz) and optical bands are being broadly explored. Communications within these bands provide massive bandwidth potential along with highly directional beam steering capabilities. While the available bandwidth offers incredible link capacity, the directionality of these technologies offers an even more significant potential for spatial capacity or area spectral efficiency. However, this directionality also implies a challenge related to the network's ability to quickly establish a connection. In this paper, we introduce a multi-tier heterogeneous (MTH) beamform management strategy that utilizes various wireless technologies in order to quickly acquire a highly directional indoor free space optical communication (FSO) link. The multi-tier design offers the high resolution of indoor FSO while the millimeter-wave (mmWave) system narrows the FSO search space. By narrowing the search space, the system relaxes the requirements of the FSO network in order to assure a practical search time. This paper introduces the necessary components of the proposed beam management strategy and provides a foundational analysis framework to demonstrate the relative impact of coverage, resolution, and steering velocity across tiers. Furthermore, an optimization analysis is used to define the top tier resolution that minimizes worst-case search time as a function of lower tier resolution and top tier range.

**Keywords:** beamforming; ultra-dense wireless networks; heterogeneous networks (HetNets); optical wireless communication (OWC); indoor free-space optics (FSO)

## 1. Introduction

The continuous growth in demand for wireless capacity projects the use of spectrum into the sub-mm, Terahertz (THz), and optical bands. Extreme directionality is a unique characteristic of THz and optical wireless communications that can be exploited in future ultra-dense networks [1–5]. Component technologies exist for short range (1–10 m) indoor free space optical communications (FSO) with steering capabilities, instantaneous coverage on the order of 1 cm$^2$, and rates exceeding 400 Gb/s [6]. Research in THz communications is driven by similar potential; however, this research area is still building momentum and is mostly dominated by photonic approaches at this time [7,8]. In general, the challenge is that directional systems require beam management to establish and maintain link connectivity for quasi-static (i.e., portable) and mobile devices. In this paper, we introduce a multi-tier heterogeneous beam management strategy that reconciles the nature of highly directional indoor FSO links and the dynamic beam management needed to maximize network performance for multiple mobile users. While this paper highlights the use of indoor FSO as the highest resolution tier, it should be noted that the modular design of the multi-tier approach could be applied to other highly directional technologies (e.g., THz communications).

We propose *a multi-tier heterogeneous (MTH) beam management strategy* that benefits from the resolution and steering velocity of various component technologies (Figure 1). The *multi-tier design* offers the high resolution of indoor FSO links while the millimeter-wave (mmWave) system narrows the FSO search space and relaxes the requirements of the FSO network in order to assure a practical time-frame to search through the device space. *Heterogeneous integration* allows for optimization of traffic distribution based on device context-of-use. This heterogeneous multi-tier design enables high resolution indoor FSO links as components of a high-performance access network serving mobile users and overcoming anomalies such as occlusions, handover, device rotation, and steering/scanning latencies.

The proposed design does not intend to compete with mmWave architectures that exist today; rather, it seeks to establish how dense networks of highly directional indoor FSO links can coexist and work collaboratively with forthcoming mmWave technologies. In addition, the MTH architecture is envisioned under the overarching coverage of a broadly available RF small cell (RFSC). This highlights the design's potential to add supplemental capacity to conventional microwave systems (e.g., WiFi). Furthermore, we envision a modular integration of the tiers within the MTH architecture. This modular design can benefit from beam management improvements at each tier and, accordingly, this manuscript does not attempt to optimize each tier individually. Rather, we consider the implicit benefit of a heterogeneous multi-tier approach to indoor FSO beam management and highlight the relative performance gains related to the coverage and resolution at each tier. In particular, we provide the following primary contributions within this manuscript:

1.  A detailed description of the MTH beam management strategy including key requirements for pointing/acquisition/tracking, multi-tier beam refinement, heterogeneous beam management, and configuration/deployment of the system
2.  A foundational analysis framework that offers a qualitative visualization of relative performance in terms of coverage, resolution, and steering velocity across tiers
3.  An optimization analysis to define the top tier resolution that minimizes worst-case search time as a function of lower tier resolution and top tier range

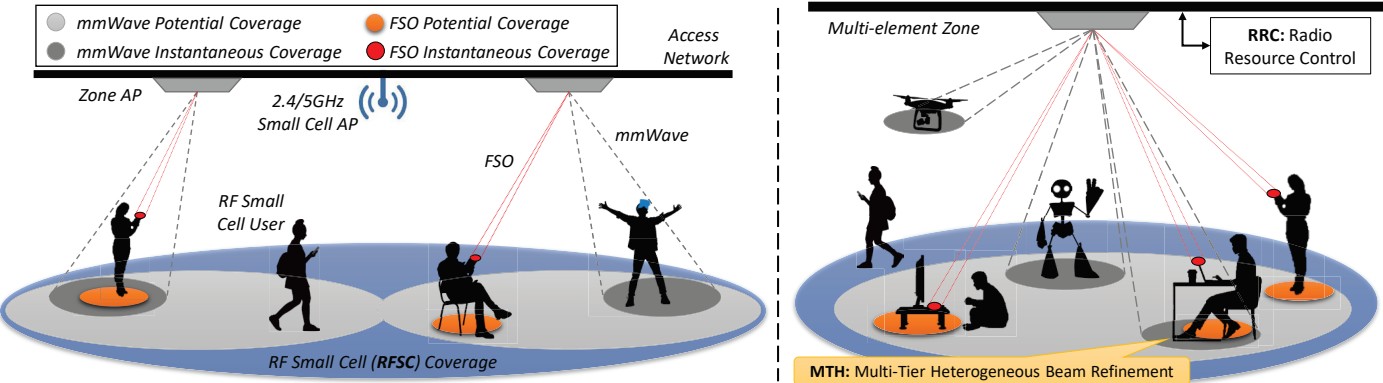

**Figure 1.** System-level view (**left**); and depiction of a single zone (**right**). *Instantaneous coverage* is defined as the coverage for a specific orientation/emission configuration (beam width, angle, etc.). *Potential coverage* is defined as the achievable coverage over all configurations.

The remainder of the paper is organized as follows. In the following section, we provide background and motivation for Indoor FSO and multi-tier heterogeneous beamforming. Section 3 then introduces the requirements of a system implementing MTH beam management. Sections 4 and 5, respectively, introduce our analysis framework and preliminary results related to optimization of the link acquisition process. Section 6 concludes the paper.

## 2. Background and Motivation

The number of mobile devices and the performance requirements of applications requiring network access continue to drive an unabated growth in wireless traffic [9]. The

increasing device density and per-device data demand continues to drive requirements for area spectral efficiency [b/s/m$^2$] or volumetric efficiency [b/s/m$^3$]. The directionality and dynamic control of steerable wireless technologies in the mmWave, THz, and optical bands offer massive potential for improvement of both area spectral efficiency and realizable system capacity [10–13]. In particular, exploiting narrow optical beams for local end-user wireless access can provide data densities and user-experienced data rates that far surpass 5G specifications. This is technically feasible based on the demonstrated potential of steered beams and the inherent signal density of focused light [6,14–20]; however, many research challenges exist at the network and system levels. These challenges must be addressed in order to realize the potential of such indoor FSO networks in practice.

Figure 2 demonstrates the breadth of characteristics for various wireless technologies and use-cases for current and future wireless devices. The left image represents the tradeoffs in directionality and coverage of various wireless technologies. Generally, the high resolution of directional communications offers the potential for massive gains in area spectral efficiency; however, the reduced instantaneous coverage also implies unique challenges when considering network provisioning for multiple dynamic users [21–23]. The right image of Figure 2 depicts the potential alignment of technologies with wireless devices that have various data demand and mobility characteristics. Applications such as interactive cloud-based services, video sourcing, augmented reality, and virtual reality have unique usage characteristics that vary greatly in terms of mobility and demand. These novel applications are changing the way that we interact with wireless networks. Furthermore, we expect unforeseen applications to contribute to the device and traffic dynamics in the coming years. This device heterogeneity motivates our vision of a heterogeneous multi-tier system, depicted in Figure 1, where the unique characteristics of various wireless technologies are optimally aligned with the characteristics of the wireless devices that they are connecting [24–26].

| | Coverage Range | Coverage Resolution | Steering Velocity |
|---|---|---|---|
| RF Small Cells (e.g., 2.4/5GHz WiFi) | + + | - - | |
| Visible Light Communication (VLC) | + | + | |
| mmWave Communications | + | + | + + |
| Fixed Free Space Optics (FSO) | - - | + + | |
| Mechanically Steered Indoor FSO | - | + + | + |
| Solid State Steerable Indoor FSO | - | + + | + + |
| Terahertz Communications | - | ++ | ++ |

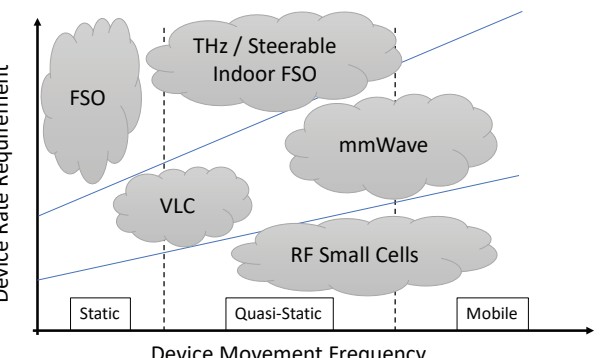

**Figure 2.** Generalized evaluation of link technology characteristics (**left**) and a context-of-use based analysis of preferred link technologies (**right**). We consider VLC as LED-based systems for both lighting and communications. FSO is broadly categorized as fixed building-to-building FSO or short range steerable FSO for indoor communications. *Range* directly related to potential coverage. *Resolution* is inversely related to instantaneous coverage. Absence of + or − indicates that the technology is typically non-steerable.

### 2.1. Background and Existing Literature

Fiber optic and FSO communications are two proven high-data-rate technologies, each with successful commercial adoption; however, both are typically used with persistent point-to-point links. Indoor optical wireless communications technologies (e.g., visible light communications or LiFi) are also beginning to see commercial acceptance; however, these systems are typically static emission systems without beam steering due to lighting requirements in the common dual-use paradigm (i.e., systems that provide both data communications and indoor illumination). Furthermore, these systems do not approach the extreme directionality of pencil-beam FSO links. Thus, novel architectures and protocols are needed at the system level in order to address the challenge of connecting mobile

and quasi-static users in dynamic environments using highly directional optical beams. At the scale of cellular networks, heterogeneous beam management techniques have been introduced in the literature in order to provide a method of coordination between macrocell and microcell base stations [27–31]. However, this heterogeneity is typically considered in the context of scale rather than wireless technology. In other words, the work referenced in [27–31] primarily considers spatial tiers of similar RF mmWave technologies that are collectively managed to avoid interference. Our vision considers the heterogeneous integration of microwave, mmWave, and Indoor FSO technologies at the scale of indoor wireless networks. Considering optimal tier alignment, mobility-aware networks have also been presented at the cellular level in order to assign resources based on mobility context [32,33]; however the scale of indoor FSO networks implies the need for novel methods that account for translational and rotational motion at higher resolution. Work in the area of heterogeneous radio and optical wireless networks has similarly introduced the idea of mobility aware or context aware resource allocation [26,34,35]; however, this has typically applied to the distribution of traffic across static RF small cells and static directional optical cells.

### 2.2. System View

Figure 1 shows our system vision where multiple user devices (UDs) receive and transmit data to overhead multi-element access points (APs) using a combination of low GHz/microwave transmission from RFSCs, mmWave directional communications, and narrow electronically steered laser sources. The mmWave APs create multiple coverage zones within the broader coverage of the RFSC. The indoor FSO links may be co-located with mmWave APs or configured as sub-zones in the potential coverage range of a mmWave AP. Each wireless technology has unique resolution and range that impacts the corresponding instantaneous and potential coverage (defined in Figures 1 and 2). Along with steering velocity, these characteristics define the time needed to scan a coverage zone. High resolution is ideal in terms of area spectral efficiency; however, resolution is inversely related to the scan time. The multi-tiered design allows for multiple levels of coverage—addressing the indoor FSO signal acquisition through beam refinement as well as providing various access tiers for UDs with different mobility traits. The radio resource control (RRC) manages the various beam configurations and resource allocation in order to bridge between the network and medium access control (MAC) layers.

### 3. MTH Beam Management Protocol

The MTH beam management protocol aims to allow multiple mobile UDs to establish and maintain a connection with fast initial access and seamless connectivity. Accordingly, the protocol must account for pointing, acquisition, and tracking (PAT) at each directional tier; heterogeneous beam refinement and handover across tiers; allocation of beams/resources in multi-UD environments; and zone configuration/deployment strategies. The protocol design must also account for physical parameters including: beam width variation, resolution, range, and transition rates at each tier; number of elements and layout of elements in each tier; and control plane feedback latency.

### 3.1. Pointing, Acquisition, and Tracking (PAT)

Each directional tier must account for the well known PAT requirements of directional communication technologies [36]. Pointing addresses the the need to direct an element's emission towards the desired UD. Pointing protocols may require the AP to scan for available UDs or use an alternative communication medium to make the system aware of the UD's presence [37–39]. After the pointing process has determined the UD's general direction, the acquisition process iteratively adapts the beam emission profile (i.e., width and direction) in order to concentrate emitted signal on the receiver. With an established connection, tracking monitors the UD's movement and potentially adapts the emission profile to maintain the link [40]. There are obvious tradeoffs between narrow and broad

emission. Wider emission simplifies tracking for small scale variations in the UD's location. Narrow emission can increase system capacity, but it requires more frequent adaptation and high speed feedback to accommodate the small scale movement.

### 3.2. Multi-Tier Beam Refinement

A high-level depiction of the beam refinement and link selection protocol for a single UD is shown in Figure 3. The three-tier beam refinement begins with the UD connecting to the RFSC and progresses through the course resolution directional tier (e.g., mmWave) and the fine-resolution indoor FSO tier. Prior to beginning the acquisition process for a higher resolution tier, the UD assesses its current mobility traits and application requirements (e.g., reliability, security, etc.) in order to make a decision about moving to the next tier. For example, a quasi-static UD that is currently in motion is likely to postpone transferring to a higher resolution tier. Similarly, a UD running a latency-constrained application may opt to remain in Tier 2 if the Tier 3 connection has intermittent outages—even if the link capacity is higher in Tier 3. Outages are more likely in Tier 3 due to the resolution of the FSO link and the fact that optical wireless communications have a higher potential for occlusions.

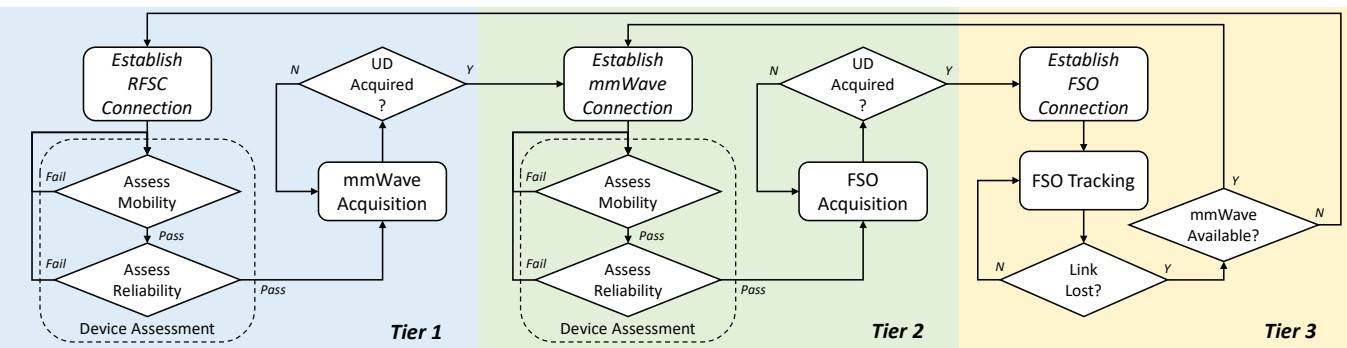

**Figure 3.** Multi-Tier Heterogeneous (MTH) beam refinement protocol. Device assessment compares UD mobility characteristics and application reliability requirements with next tier characteristics prior to acquiring a higher resolution link.

### 3.3. Beam Management

A basic instantiation of the MTH protocol can assume that individual UDs operate with a greedy approach and attempt to connect to the tier with highest resolution and link capacity; however, UD mobility, reliability/security requirements, and other device characteristics can also be considered when determining the tier that maximizes average throughput. An extension of the single-UD protocol would account for aggregate system capacity and resource allocation across multiple UDs. To couple the beam management across tiers for both single-UD beam refinement and multi-UD network configuration, the MTH beam management protocol should sit within the radio resource control (RRC) between the Network/Internet layer and the Link layer (Figure 4). This abstract view of the protocol implementation demonstrates the importance of tight integration with both the Data link/Physical layers and the Network layer. The protocol must integrate with the Data link/Physical layers in order to manage the PAT process across tiers. It should also integrate with the Network layer in order to manage routing of data traffic across tiers during beam refinement and/or accommodate resource allocation decisions in scenarios where the optimal tier is related to the UD context-of-use.

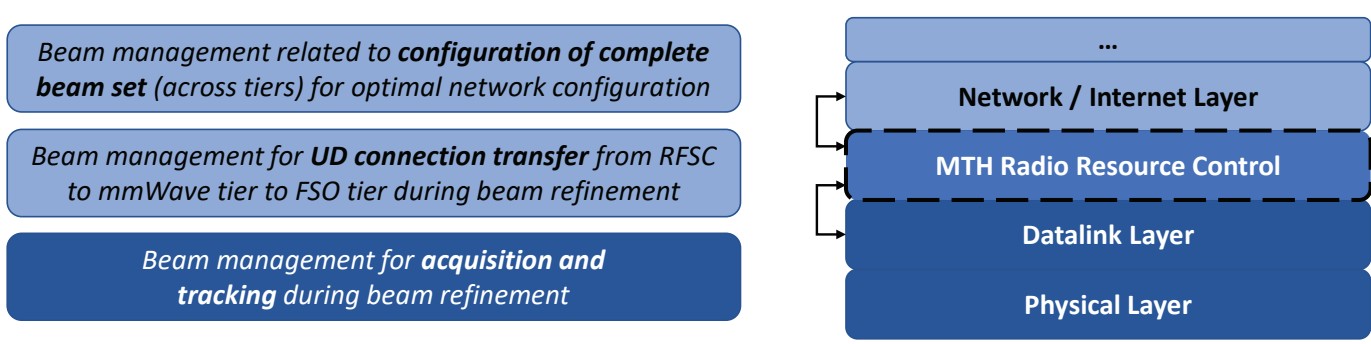

**Figure 4.** MTH radio resource control for beam management in the TCP/IP 5-layer model.

### 3.4. Zone Configuration

Optimal system deployment should account for zone configuration—including the number of elements per tier and the coverage associated with each element. The obvious motivation for multiple elements is to provide an opportunity for multiple simultaneous links; however, multiple elements also impact coverage and redundancy. Providing Tier 3 access throughout the Tier 2 zone requires a single element to steer to any location within the zone. To relax this requirement, the indoor FSO network may be deployed to only cover a portion of the zone or, alternatively, be configured with sectors or subzones assigned to different elements (Figure 5). The challenge with sectors is that the protocols must account for additional transitions as UDs move between sectors (i.e., from subzone to subzone). Distributed elements also add redundancy and mitigate outage concerns. When elements are distributed, potential coverage regions can overlap without leading to contention due to the small instantaneous coverage. This provides redundancy in the potential links for a UD to connect to, as we demonstrated in [19].

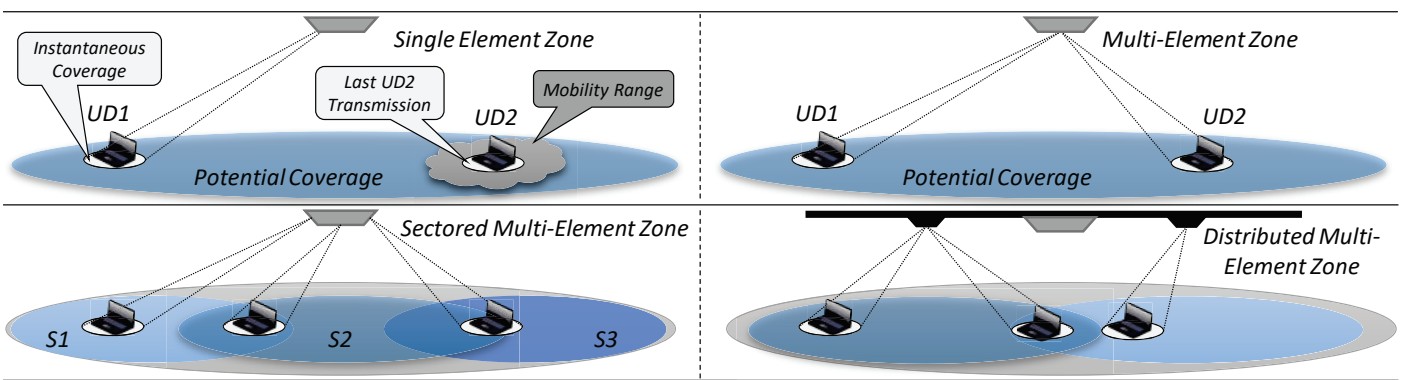

**Figure 5.** Potential coverage, instantaneous coverage, and associated multiple access strategies for single element (**top left**), multi-element (**top right**), and sectored multi-element (**bottom left**) steerable FSO links as well as a sectored multi-element zone with distributed elements (**bottom right**).

### 4. Analysis Framework

In this paper, we highlight a specific focus on the optimization of the beam refinement process. We aim to allow UDs to connect to the highest resolution/maximum throughput tier while minimizing access delay — both during initial access and upon occurrence of an outage. The heterogeneous beam refinement allows the Tier 2 search to occur while other UDs use Tier 3 for communication. Furthermore, a variety of search techniques (RSSI, ToF, AoA, etc.) may be used, and the protocol may avoid connecting to the middle tiers during beam refinement in order to speed up access to the higher tier.

Our preliminary analysis focuses on the ideal relationship between range (i.e., potential coverage), resolution, and steering velocity at each tier. To demonstrate the optimization problem, consider a three-tier beam refinement strategy where each tier narrows the search space to the area covered by a single configuration, and the next tier searches over the

space covered by a single configuration of the previous tier. Assuming non-overlapping configurations, the worst case search time is

$$T_s = T_d + t_2 n_2 + t_3 n_3 = T_d + t_2 \alpha_2 \beta_2 + t_3 \alpha_3 \beta_3 \tag{1}$$

where $T_d$ is the RFSC's device discovery time, $t_i$ is the time to switch between configurations for tier $i$, and $n_i$ is the number of configurations in the $i$th tier. With the assumption of non-overlapping configurations, we can define $n_i = \alpha_i \beta_i$ where $\alpha_i$ and $\beta_i$ are the resolution and range of tier $i$, respectively. We define the tier resolution as the number of configurations per unit area and the range as the total area covered by the tier. These parameters are summarized in Table 1.

**Table 1.** MTH analysis parameters.

| Parameter | Description |
|-----------|-------------|
| $T_s$ | Maximum Search Time |
| $T_d$ | Discovery Time (Tier 1) |
| $t_i$ | Configuration Switching Time (Tier $i$) |
| $n_i$ | Number of Configurations (Tier $i$) |
| $\alpha_i$ | Tier $i$ Resolution [configurations/$m^2$] |
| $\beta_i$ | Tier $i$ Range [$m^2$] |

Assuming that the search space of tier $i + 1$ is equal to the instantaneous coverage (i.e., area covered by a single configuration) of tier $i$, then $\beta_{i+1} = 1/\alpha_i$. In other words, tier $i$ narrows the search space to an area covered by a single configuration and tier $i + 1$ searches this space. Accordingly, Equation (1) can be rewritten as either of the following:

$$T_s = T_d + t_2 \alpha_2 \beta_2 + t_3 \frac{\alpha_3}{\alpha_2} \tag{2}$$

$$T_s = T_d + t_2 \frac{\beta_2}{\beta_3} + t_3 \alpha_3 \beta_3 \tag{3}$$

The key observation in this analysis is the impact of $\alpha_2$ (or, similarly, $\beta_3$). Specifically, there is an ideal resolution of Tier 2 that minimizes $T_s$ and this resolution is dependent on the range of Tier 2 and resolution of Tier 3. Furthermore, the relative importance of $\beta_2$ and $\alpha_3$ is dependent on the relative switching speeds of Tiers 2 and 3. To determine the optimal Tier 2 resolution, we must determine $\alpha_2$ that minimizes $T_s$ for given $T_d$, $t_2$, $t_3$, $\beta_2$ and $\alpha_3$. Observing Equations (2) and (3), we can find the rate of change of $T_s$ with respect to $\alpha_2$ and $\beta_3$, respectively.

$$\frac{d}{d\alpha_2} T_s = t_2 \beta_2 - t_3 \frac{\alpha_3}{\alpha_2^2} \tag{4}$$

$$\frac{d}{d\beta_3} T_s = -t_2 \frac{\beta_2}{\beta_3^2} + t_3 \alpha_3 \tag{5}$$

Given that $T_s$ is a convex function for $\alpha_2 \in \mathbb{R}_{>0}$, we can set the derivative from Equation (4) to zero and solve for the positive value of $\alpha_2$ (i.e., Tier 2 resolution) in order to find the value of $\alpha_2$ that minimizes $T_s$. We can similarly observe Equation (5) in order to view the impact in terms of $\beta_3$ (i.e., Tier 3 range or, similarly, Tier 2 instantaneous coverage). Following this process, we find the optimal Tier 2 resolution and Tier 3 range in Equations (6) and (7), respectively. Given that $\beta_{i+1} = 1/\alpha_i$ for this specific problem formulation, we recognize the direct relationship between the optimal Tier 2 resolution and Tier 3 range. As such, the optimal value for $\beta_3$ could be directly derived from knowledge of $\alpha_2$ (or vice versa). However, the inverse relationship of the derived results in Equations (6) and (7) demonstrates that the general relationship holds and offers a sense of validation. Furthermore, the direct relationship implies that all relevant information

can be defined by either $\alpha_2$ or $\beta_3$; however, each can be used to better visualize the relative performance impact under different operating conditions (as seen in Section 5).

$$\underset{\alpha_2 \in \mathbb{R}_{>0}}{\arg\min}(T_s) = \sqrt{\frac{t_3\alpha_3}{t_2\beta_2}} \tag{6}$$

$$\underset{\beta_3 \in \mathbb{R}_{>0}}{\arg\min}(T_s) = \sqrt{\frac{t_2\beta_2}{t_3\alpha_3}} \tag{7}$$

Under the given assumptions, these results could be used to define the ideal mmWave (Tier 2) emission profile for a system with known FSO emission profile and FSO/mmWave configuration switching times along with a specified coverage area. However, these simplifying assumptions are obviously limited in terms of practical application since a more realistic system would account for configuration overlap and non-uniform coverage, feedback and pipelined scanning, and other protocol characteristics. Accordingly, these results are intended primarily for qualitative analysis in order to observe the relative impact of various system parameters.

## 5. Results and Analysis

To observe the relative impact of the parameters, we evaluate the search time for a few scenarios where the switching time of the Tier 3 FSO link is faster than, equivalent to, or slower than the switching time of the Tier 2 mmWave link. To provide a more general analysis, all results are depicted in time relative to the switching time of the mmWave link (i.e., $t_2$). Without loss of generality, we also assume $T_d = 0$ since the Tier 1 discovery time is a constant offset independent of the other parameters.

The first results, depicted in Figure 6, illustrate the search time when the FSO tier can switch between configurations much faster than the mmWave tier (i.e., $t_2 = 100t_3$). The plot on the left shows the search time versus $\alpha_2$ and the plot on the right shows the corresponding results as a function of $\beta_3$. In each plot, results are shown for combinations of $\beta_2 \in \{9, 25\}$ and $\alpha_3 \in \{625, 2500, 10,000\}$. These Tier 3 resolution values represent FSO links with instantaneous coverage of 16, 4, and 1 cm$^2$, respectively. When the Tier 2 resolution is low, we can see that the search time is heavily influenced by the Tier 3 resolution (i.e., the third component of Equation (2)). This implies that the Tier 2 scan completes quickly and most of the time is spent searching within Tier 3. Conversely, as the resolution increases there are more Tier 2 configurations in a given space and fewer Tier 3 configurations within the instantaneous coverage of a single Tier 2 configuration; therefore, the scan spends most of its time in Tier 2. Accordingly, the search time is heavily influenced by the Tier 2 range (i.e., the second component of Equation (2)). This is similarly seen in the right plot where search time is heavily influenced by Tier 2 range when Tier 3 range is small and heavily influenced by Tier 3 resolution when Tier 3 range is large.

In Figure 7, we demonstrate the impact of relative switching times—specifically, we show results for scenarios where: (a) switching between FSO configurations is 10× faster than switching between mmWave configurations; (b) FSO and mmWave switching times are equivalent; and (c) switching between FSO configurations is 10× slower than switching between mmWave configurations. The key observation here is that the trends are similar; however, increasing the time to switch between FSO configurations or reducing the time to switch between mmWave configurations shifts the optimal Tier 2 resolution to the right and, accordingly, reduces the optimal Tier 3 range. Intuitively, this makes sense since reducing the relative performance of Tier 3 implies that the system should increase the utilization of Tier 2.

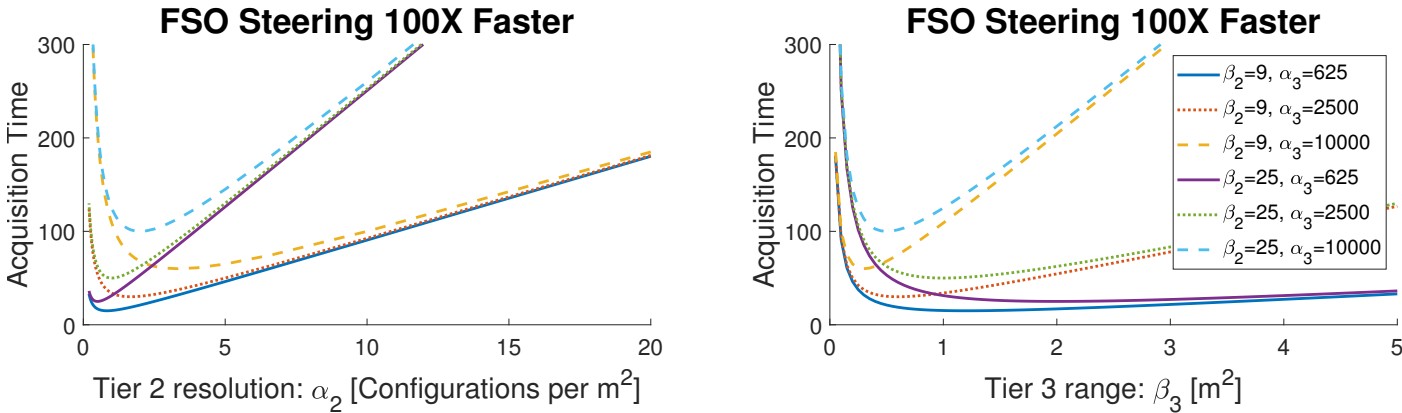

**Figure 6.** Maximum search time for MTH protocol as a function of Tier 2 resolution (i.e., $\alpha_2$) and Tier 3 range (i.e., $\beta_3$) when $t_2/t_3 = 100$.

**Figure 7.** Maximum search time for MTH protocol as a function of Tier 2 resolution (i.e., $\alpha_2$) and Tier 3 range (i.e., $\beta_3$). Results are shown for: $t_2/t_3 = 10$ (**top**); $t_2 = t_3$ (**middle**); and $t_2/t_3 = 0.1$ (**bottom**). The legend in the bottom right plot is common to all plots.

Lastly, Figure 8 highlights the impact that Tier 2 range, Tier 3 resolution, and relative switching times have on the optimal Tier 2 resolution (or, similarly, Tier 3 coverage). The obvious observation is that the three sets of figures appear similar; however, the color bar range shows that relative switching time has an impact on magnitude of the outcomes. As FSO switching becomes faster, Tier 3 can explore a larger space in a similar time, therefore the optimal Tier 2 resolution (i.e., $\alpha_2$) decreases and the optimal Tier 3 range (i.e., $\beta_3$) increases. Observing the impact of $\beta_2$ and $\alpha_3$, we can see that larger spaces (i.e., larger $\beta_2$ values) increase the optimal Tier 3 range in order to distribute the additional required search time across both tiers. Increased Tier 3 resolution drives the FSO link towards pencil-beam emission, requiring more Tier 3 configurations over a given area; thus, the optimal Tier 2 resolution increases to again better utilize Tier 2 in the search.

To reiterate the point from above, this analysis presents a qualitative example of the tradeoffs; however, quantitative analysis depends on the uniformity of coverage, hardware characteristics (e.g., steering velocity), and the feedback latency of the control messaging protocol between AP and UD, which may be pipelined. Accordingly, this analysis demonstrates tradeoffs in design parameters, but the optimal search time ultimately depends on the protocol characteristics.

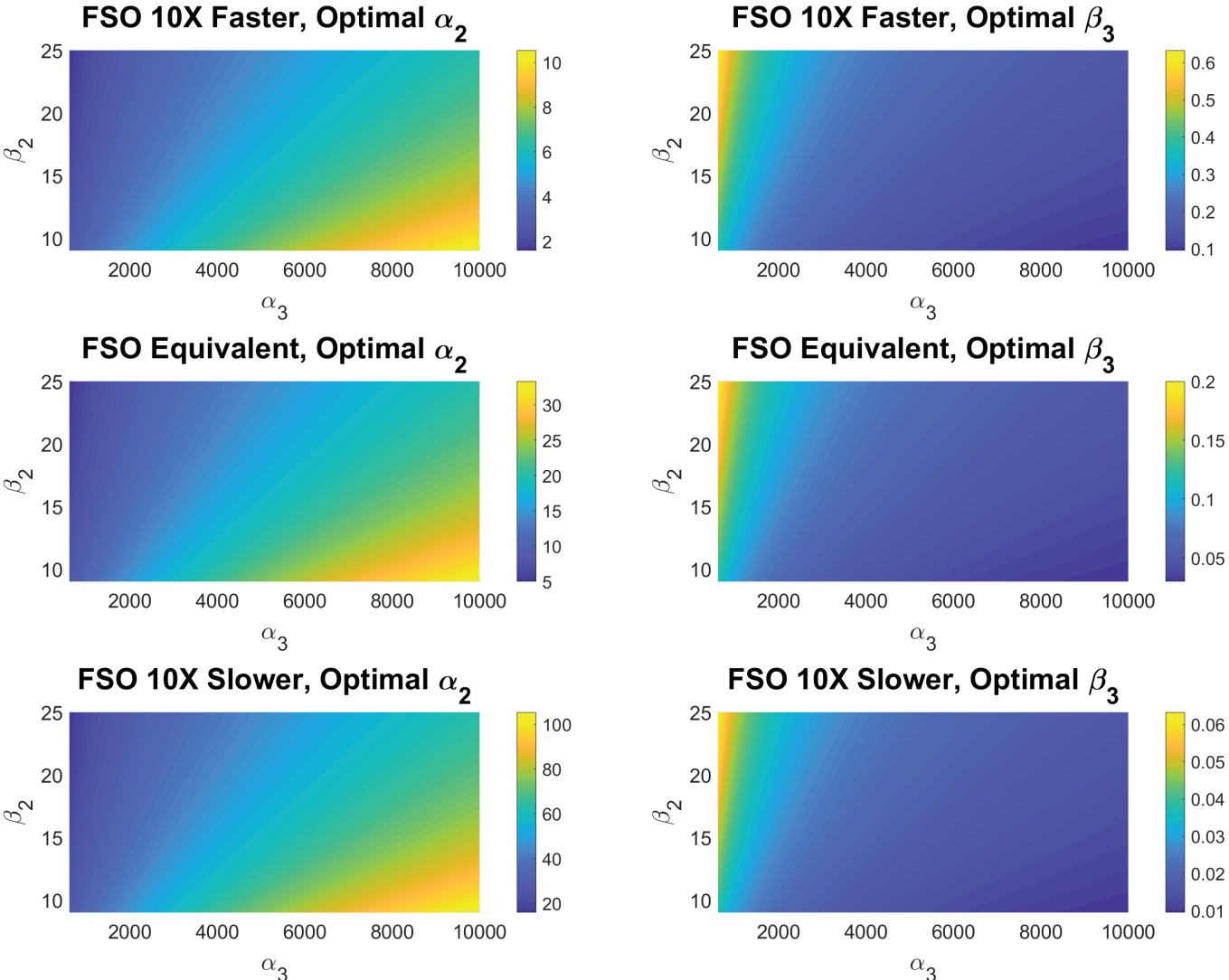

**Figure 8.** Optimal Tier 2 resolution (i.e., $\alpha_2$) and Tier 3 range (i.e., $\beta_3$) for systems where relative switching times are related by: $t_2/t_3 = 10$ (**top**); $t_2 = t_3$ (**middle**); and $t_2/t_3 = 0.1$ (**bottom**).

## 6. Conclusions

In this paper, we provide a thorough overview of the requirements and considerations for a novel multi-tiered heterogeneous beam management strategy that promises to: (a) improve traffic distribution across technologies; and (b) improve initial access time of a steerable indoor FSO network. In particular, we introduce a high level beam refinement protocol that accounts for device characteristics and link reliability before moving to a tier with higher directionality. The protocol also aims to improve indoor FSO acquisition time by taking advantage of the unique characteristics of directional technologies operating at different tiers. To evaluate the relative impact of various system parameters, we developed a qualitative analysis framework for the proposed MTH beam management strategy. This framework was used to demonstrate tradeoffs in the design decisions and derive an optimization analysis that minimizes the worst-case search time in a three tier implementation. While the presented evaluation is intended primarily for relative comparison, the analysis motivates a tighter coupling and integration of various directional communication technologies within the future wireless communications ecosystem.

**Author Contributions:** Conceptualization, M.B.R., T.D.C.L. and M.H.; Formal analysis, M.B.R.; Writing—original draft, M.B.R.; Writing—review & editing, T.D.C.L. and M.H. All authors have read and agreed to the published version of the manuscript.

**Funding:** This research received no external funding.

**Data Availability Statement:** The data presented in this study are available on request from the corresponding author.

**Conflicts of Interest:** The authors declare no conflict of interest.

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
