# Peer review of "Multi-Tier Heterogeneous Beam Management for Future Indoor FSO Networks"

_applsci, doi:10.3390/app11083627_

Round 1

Reviewer 1 Report

The use of THz sources  is problematic due to the limited power levels achievable with the present technolgy. the authers need to discuss, how they plan to overcome this.

Why do they avoid microwave frequencies, since here better signal level generation is available. It appears to the referee, they the authors are intending to bring intop this discussion the THz concept because they have the impression that the reference to THz waves is a new concept. This is a debatable issue.

Reviewer 2 Report

Hereafter all my suggestions and considerations:

1) Line 16: fix "teirs" to "tiers"

2) Section 2: the background section does not provide enough details about similar systems already present in literature;

3) Figure 4 label: fix "TC/IP" to "TCP/IP";

4) Can you clarify why you would put the "MTH RRC" layer between Datalink and Network Layers"?

5) Line 203: you have never defined t1;

6) Equations 6 and 7: the 2 quantities are stricly related, I do not understand why you performed calculations for both of them;

I would like to conclude my review saying the general idea of the paper is good and can lead to very interesting considerations, but it has been merely reduced to a simple maximization exercise since there are no comparisons with other beam management systems. It is difficult for readers to understand why the solution you proposed is improving the signal acquisition time.

Round 2

Reviewer 1 Report

Your improvements are good.

Reviewer 2 Report

Authors have made an effort to increase the quality of the work and the paper has been quite improved.